# Designing Sustainable Polymer Blends: Tailoring Mechanical Properties and Degradation Behaviour in PHB/PLA/PCL Blends in a Seawater Environment

**DOI:** 10.3390/polym15132874

**Published:** 2023-06-29

**Authors:** Leonardo G. Engler, Naiara C. Farias, Janaina S. Crespo, Noel M. Gately, Ian Major, Romina Pezzoli, Declan M. Devine

**Affiliations:** 1PRISM Research Institute, Technological University of the Shannon: Midlands Midwest, Athlone Campus, University Road, N37 HD68 Athlone, Ireland; a00278634@student.tus.ie (L.G.E.);; 2Postgraduate Program in Materials Science and Engineering, University of Caxias do Sul, Francisco Getúlio Vargas Street, 1130, Caxias do Sul 95070-560, Brazil; 3Applied Polymer Technologies Gateway, Technological University of the Shannon: Midlands Midwest, Athlone Campus, University Road, N37 HD68 Athlone, Ireland

**Keywords:** poly(lactic acid) (PLA), poly(hydroxybutyrate) (PHB), polycaprolactone (PCL), thermal degradation, seawater degradation, biodegradable blends

## Abstract

Biodegradable polyesters are a popular choice for both packaging and medical device manufacture owing to their ability to break down into harmless components once they have completed their function. However, commonly used polyesters such as poly(hydroxybutyrate) (PHB), poly(lactic acid) (PLA), and polycaprolactone (PCL), while readily available and have a relatively low price compared to other biodegradable polyesters, do not meet the degradation profiles required for many applications. As such, this study aimed to determine if the mechanical and degradation properties of biodegradable polymers could be tailored by blending different polymers. The seawater degradation mechanisms were evaluated, revealing surface erosion and bulk degradation in the blends. The extent of degradation was found to be dependent on the specific chemical composition of the polymer and the blend ratio, with degradation occurring via hydrolytic, enzymatic, oxidative, or physical pathways. PLA presents the highest tensile strength (67 MPa); the addition of PHB and PCL increased the flexibility of the samples; however, the tensile strength reduced to 25.5 and 18 MPa for the blends 30/50/20 and 50/25/25, respectively. Additionally, PCL presented weight loss of up to 10 wt.% and PHB of up to 6 wt.%; the seawater degradation in the blends occurs by bulk and surface erosion. The blending process facilitated the flexibility of the blends, enabling their use in diverse industrial applications such as medical devices and packaging. The proposed methodology produced biodegradable blends with tailored properties within a seawater environment. Additionally, further tests that fully track the biodegradation process should be put in place; incorporating compatibilizers might promote the miscibility of different polymers, improving their mechanical properties and biodegradability.

## 1. Introduction

As our dependence on plastics continues to grow globally, the amount of plastic waste that finds its way into our oceans, rivers, and landfills is largely a result of human behaviour. This trend has catastrophic consequences, as plastic particles are now being incorporated into the food chain of animals, including humans, with unpredictable consequences [1,2]. Given that oceans account for approximately 97% of the world’s water supply and cover nearly 70% of the Earth’s surface, it is likely that a significant portion of this plastic waste ultimately ends up in our oceans, particularly when not properly disposed of [3].

Biodegradable polymers are a viable approach to addressing the issue of plastic pollution in marine ecosystems. In contrast to conventional plastics, these polymers are capable of rapidly decomposing over a brief span of a few months through natural mechanisms, such as enzymatic digestion, hydrolysis, oxidation, or mechanical degradation, and are hence environmentally friendly. Nevertheless, not all biodegradable polymers degrade in seawater; for instance, PHB and PCL are known to biodegrade in seawater [4,5], whereas PLA is not [6,7]. Therefore, it is crucial to study the blending process and characterisation of these polymers to enable the comprehensive seawater degradation of these materials.

Seawater solutions contain a higher content of salts than freshwater, with chloride, sodium, sulphate, magnesium, calcium, and potassium making up 99% of the ionic salts present [8]. The temperature of seawater varies between 30 °C on the surface in summer and −1 °C on the seabed during winter, depending on the season and geographical region, while the pH can range from 7.5 to 8.4 [1,9]. Additionally, previous studies have shown that the natural dynamic movement of waves in oceans can promote degradation by mechanical fragmentation of the biodegradable polymers studied, such as PLA, PCL, and PHAs [6,10].

Reproducible investigations of seawater degradation in the natural environment are challenging owing to the varying climatic conditions in different regions. However, laboratory experiments can standardise the experimental parameters, increasing the likelihood of reproducible results [1]. Similarly, when blending PLA with more flexible materials, such as PCL and PHB, the goal is to enhance flexibility while preserving the mechanical properties of the blends. Although this approach shows promise, it is essential to overcome the limitations of multi-material mixtures using techniques such as plasticisation, copolymerisation, and blending [11,12]. The use of different polymers to produce ternary blends is an interesting alternative that can be used to tailor the desired properties of materials. In the context of biomedical polymeric devices, understanding their degradation process is crucial for ensuring their stability and functionality. By examining the degradation mechanisms and behaviour of these ternary blends, researchers can gain valuable insights into how the materials perform under real-world conditions. This knowledge enables the development of more robust and reliable biomedical devices that can withstand the challenges of the human body and contribute to improved patient outcomes.

The degradation process of biomedical polymeric devices is a multifaceted phenomenon that is subject to variations in the environmental conditions and kinetics, which depend on the processing settings, temperature, sample geometry, and packing properties; e.g., moulded parts are more densely packed than extruded components, which are susceptible to die swelling (the expansion due to polymer chain realignment) once they exit the extruder [13,14,15], therefore presenting a more complex degradation process than in seawater. Similarly, the degradation of such medical devices can also occur via four distinct mechanisms, namely hydrolytic, oxidative, enzymatic, and physical degradation [1]. Hydrolytic degradation is triggered by the interaction between the polymeric chains and the water present in the surrounding tissues. On the other hand, oxidative degradation results from the release of oxidants by tissues, which acts as a biological defence mechanism against the implant. Enzymatic degradation also involves a biological response, but it varies from patient to patient and from tissue to tissue over time. Finally, physical degradation occurs as a result of water swelling and mechanical tension, leading to changes in the glass transition temperature and geometry of the swollen polymers. This, in turn, affects the mechanical properties of the device and could potentially lead to fractures, particularly in situations where friction is involved in the motion under pressure [1,16].

Narancic et al. (2018) [15] previously examined the marine biodegradability of 25 mg extruded samples of different biodegradable polymers and found that only PHB and thermoplastic starch (TPS) were marine degradable. To overcome this challenge, the novelty of this work is to determine whether, through the production of ternary blends containing common biodegradable polymers, PLA, and PCL, a commercially relevant marine biodegradable polymer blend could be produced. Furthermore, most medical devices and packaging types are produced by injection moulding which manufactures parts that have tightly packed polymer chains compared to extruded parts and are more challenging from a biodegradation point of view.

The selection of polymer concentrations in ternary blends of PHB, PLA, and PCL, is also challenging and can be justified based on various factors. Firstly, each polymer in the blend contributes distinct properties, with PHB offering biodegradability and excellent mechanical properties, PLA providing stiffness and strength, and PCL imparting flexibility and toughness to the blends [13,14]. Accordingly, this study proposes that the low flexibility of PLA can be mitigated by blending it with more pliable materials such as PCL and PHB through a blending process, while simultaneously tailoring its degradation properties. The selected blends exhibit enhanced flexibility and can be applied in various fields, including packaging and medical devices, particularly in the production of biodegradable straws as an alternative to single-use plastics. Additionally, optimal blends will serve in future studies as feedstock material for the development of flexible biodegradable ureteral stents.

## 2. Materials and Methods

### 2.1. Materials

Poly-(lactic acid) (PLA 4043D) was purchased from NatureWorks (Minnetonka, MN, USA), it has a melt flow index (MFI) of 4.9 g·10 min^−1^, and a density of 1.24 g·cm^−3^. Poly-(3-hydroxybutyrate) (PHB P226) was purchased from Biomer (Krailling, Germany), it is highly crystalline with an MFI of 16.3 g·10 min^−1^ and a density of 1.25 g·cm^−3^. Polycaprolactone (PCL CAPA 6500) was purchased from Ravago Chemicals (Barnsley, UK), it has a high molecular weight (M_w_ 50,000 g·mol^−1^), and MFI of 7.0 g·10 min^−1^, all MFI tests were performed at a temperature of 175 °C and a load of 2.16 kg, following standard methods. All polymers were used as received and they were dried with forced hot air circulation at 45 °C for 24 h before processing, to avoid any moisture residue that could promote the hydrolysis degradation of the polyesters.

### 2.2. Material Extrusion

The compounding of materials was carried out on a Leistritz Micro 27 twin-screw co-rotating extruder (Leistritz Group, Nuremberg, Germany) with a 27 mm screw diameter and a 38:1 length-to-diameter ratio, with a constant screw speed of 150 rpm and temperature profile from feeder to die of 110/120/130/140/150/160/170/180 °C, and two additional heating zones at the head of the extruder, a flange and a strand die, at 190 and 200 °C, respectively. The extruder was fed at 4 kg·h^−1^, and the compounded samples were named as PHB, PLA, PCL, 50/50/0, 50/0/50, 50/25/25 and 30/50/20 (wt.% PHB/PLA/PCL). The filament produced was pelletised to 3 mm granulates after cooling down in a conveyor belt.

### 2.3. Material Moulding

Tensile specimens with dimensions (type V): length 63.50 mm, width 9.50 mm, gauge length 9.50 mm, gauge width 3.18 mm, and thickness 3.20 mm, were moulded according to ASTM D638 on a BabyPlast^®^ 6/12 (Rambaldi, Italy) injection moulding process, equipped with a 14 mm diameter piston. The machine possesses three temperature-controlled zones: plasticising zone, chamber and nozzle represented as zone 1, 2 and 3, respectively. The injection moulding parameters were optimised for each sample and are described in Table 1. At least 5 specimens for each sample were produced.

### 2.4. Seawater Degradation

For the seawater degradation study, a marine broth solution based on ASTM D6691 was prepared and placed into a 22 L unstirred thermostatic bath Clifton^TM^ (Oldmixon Cres, Weston, UK), allowing the temperature constant to maintain at 30 °C for up to 8 weeks of the experimental testing. The marine broth was prepared by weighing all the individual components of the solution in a high precision analytical scale (±0.1 mg), before dissolving them in distilled water under constant stirring for 3 h to promote solution homogeneity. A commercial fish tank pump was introduced to the thermostatic bath to promote the aeration of the solution and prevent components of the solution from depositing at the bottom of the bath.

Moulded tensile samples were weighed and placed into the seawater solution distanced at a minimum of 2 cm from each other, to assess degradation during 0, 14, 28, and 56 days with 3 minimum replicates for each sample. Each collected sample was washed and dried before testing.

### 2.5. Weight Variance

The weight variance of samples was measured from the beginning of the degradation tests and after 14, 28, and 56 days of seawater degradation. After each collection date, the samples were removed from the seawater environment, washed with current water, and dried for 24 h in an oven at 40 °C. Their final weight was measured and recorded using a Sartorius Entris analytical balance (Sartorius AG, Göttingen, Germany), prior to any subsequent testing.

### 2.6. Mechanical Properties

Tensile testing was conducted in accordance with ASTM D3039 standard procedure using a ZwickRoell Z010 universal testing machine (ZwickRoell Ltd., Ulm, Germany) equipped with a 5 kN load cell. The tests were carried out at a strain rate of 5 mm·min^−1^, at room temperature and the tensile properties were obtained directly from the equipment analysis software. A minimum of 3 replicates were evaluated per group, and the dimensions of each sample were recorded prior to testing. The stress–strain curves were obtained from the software TestXpert II (ZwickRoell, Ulm, Germany), and utilised for the calculation of the Young Modulus.

### 2.7. Infrared Analysis

The infrared analysis was performed using a Spectrum One (Perkin Elmer, Waltham, MA, USA) with a universal attenuated total reflectance (ATR) sample adapter. All data were obtained at room temperature (20 °C) in the spectral range between 4000–650 cm^−1^ against a background of air, using 10 scans per sample cycle and a resolution of 4 cm^−1^. The compression force was kept constant at 80 N and the data were collected from the spectrum software used to perform the analysis. The chemical groups were evaluated in regions by performing the ratio area of the groups related to the main chain (−CH3 and −CH2) and the oxygenated groups (–C=O, –C–O). The areas of –CH3, –CH2 stretches; A_1_ (3050–2800 cm^−1^); –C=O: A_2_ (1840–1600 cm^−1^); and ––O: A_3_ (1250–1000 cm^−1^) were determined using the software Origin (OriginLab^©^). Results were expressed as the mean and standard deviation.

### 2.8. Thermal Analysis

By using a thermal analyser (TGA-50, Shimadzu Co., Kyoto, Japan) with a dynamic nitrogen atmosphere of 50 mL·min^−1^, the blends were characterised by thermogravimetric analysis to evaluate the thermal stability of the blends produced. Samples weighing between 8–12 mg were heated from room temperature (20 °C) to 700 °C at a rate of 10 °C·min^−1^, providing information about the weight loss plotted against the temperature for each sample. Additionally, a derivative curve (DTG) was produced to indicate the temperatures at which maximum rates of weight loss occurred.

The thermal properties of the samples were investigated in a differential scanning calorimeter DSC Pyris 4000 (Perkin Elmer, Waltham, MA, USA). Samples weighing between 4–6 mg were crimped in non-perforated aluminium pans with reference to an empty crimped pan. The analysis was carried out with a nitrogen flow rate of 30 mL·min^−1^ to prevent the sample degradation. The samples were heated from −60 to 190 °C at a rate of 10 °C·min^−1^ holding the temperature for 2 min before cooling down to −60 °C and holding the temperature for 2 min again, this process is repeated twice for each sample. The thermal properties of the samples were analysed with the equipment software Pyris Series DSC 4000 (Perkin Elmer, Waltham, MA, USA) and the data analysed based on the second heating cycle of the test. The crystallinity of the samples was calculated with the values in Table 4 by applying the crystallinity index equation,
(1)$$C_{I}=\frac{\Delta {\mathrm{Hm}}_{\;}}{\Delta {\mathrm{Hm}}_{0}\times f}\times 100\%$$
where C_I_ is the crystallinity index, ∆Hm is the experimental melting enthalpy, f is the polymer weight fraction and ∆Hm_0_ is the theoretical melting enthalpy of the pure material (considering it as 100% crystalline). The theoretical melting enthalpy for PHB, PLA and PCL are 146 J·g^−1^, 93.7 J·g^−1^, and 135 J·g^−1^, respectively [17,18,19].

Dynamic mechanical thermal analysis (DMTA) was run to provide information about thermal transitions (T_g_ values) that were not possible to identify in the previous thermal analysis, due to the overlapping of these properties in the blends of this study. DMTA was carried out in single-cantilever mode using a Perkin Elmer (Waltham, MA, USA) DMA 8000. Rectangular samples of size 20 × 10 × 3 mm^3^ were subjected to a temperature ramp from −80 up to 100 °C at a constant heating rate of 3 °C·min^−1^. A frequency of 1 Hz was used for all tests and the maximum flexural deformation (γ) was set at 10 µm.

### 2.9. Morphology Characterisation

Tensile samples were subjected to a cryofracture process using liquid nitrogen to observe the internal morphology of the samples without any deformation in the inner layers of the sample. A field emission scanning electron microscope (Tescan Mira, Oxford Instruments, Cambridge, UK) (5 kV) was used to assess the degradation areas of samples after 56 days in the seawater environment.

## 3. Results and Discussion

### 3.1. Seawater Degradation

As reported in previous studies [20], the degradation of a biomaterial may occur through four different mechanisms: hydrolytic degradation (scission of polymeric chains by hydrolytic activity), enzymatic degradation (scission of polymeric chains by enzymatic activity), oxidative degradation (radical attack supported by peroxide-producing inflammatory reactions, when in vivo), and physical degradation (depending on the physical activity of the environment, swelling-deswelling, etc.) [21].

Surface erosion and bulk degradation represent two distinct mechanisms of hydrolytic degradation, although their occurrence may not always be mutually exclusive [22]. Bulk degradation is the predominant mechanism of degradation in most polyester-based polymers, and when the rate of water diffusion is greater than the rate of hydrolysis (Dt >> Ht), sample saturation by the degradation media and a non-linear mass loss over time occur [20]. Additionally, as a result of the accumulation of degradation products, it is common to have autocatalytic reactions in situ, which leads to areas of accelerated degradation that might compromise or cause the structure to collapse [14]. Surface erosion, on the other hand, commonly presents a linear loss of mass over time and is characterised by a faster rate of hydrolysis than water diffusion (Dt ≤ Ht). This effect reduces the accumulation of degradation products since they are free to diffuse away from the polymeric matrix; therefore, no autocatalytic reactions occur [20].

The phenomenon of bulk degradation in poly(α-hydroxy-esters) samples with a thickness (Lc) up to 7.4 cm was previously reported in studies with temperatures ranging between 15 and 37 °C, which reported an increased rate of PHA and PLA degradation in alkali environments. Additionally, the onset of percent crystallinity change during hydrolysis was observed at approximately 37 °C, corresponding to the temperature at which the hydrolysis of PLA chains starts to occur noticeably [22,23,24,25]. This has been demonstrated through experimental results as well, as depicted in Figure 1. Over a period of 56 days, an observable increase in the weight of some samples is noted, attributable to the formation of fissures and cracks, which permit the infiltration of aqueous solutions into the PLA-based samples, leading to a transient rise in their mass, as observed in samples of 100 wt.% of PLA, 50/50/0, and 30/50/20 (wt.% PHB/PLA/PCL). The same investigations also demonstrated that microbial activity exerts a marginal to negligible impact on the hydrolysis phase of degradation, which involves the cleavage of ester linkages in PLA samples [24]. PHB and PCL samples exhibited notable mass variations throughout the 56 days of seawater degradation, whereby PCL demonstrated a weight reduction of up to 10%. Furthermore, the specimens containing higher amounts of these two polymers exhibited comparable behaviours, as observed for the samples 50/0/50 and 50/25/25 (wt.% PHB, PLA, and PCL) in Figure 1. It is assumed that surface erosion serves as the main degradation mechanism for these samples, as evidenced by the constant decrease in weight over time. Bagheri et al. (2017) [7] investigated the biodegradability of PHB, PLA, PCL, and poly(lactic-co-glycolic acid) (PLGA) in a seawater environment at a constant temperature of 25 °C for over a year; only the latter underwent complete degradation within this timeframe, while the other polymers exhibited minimal degradation. Specifically, PHB demonstrated only 6% weight loss, whereas PLA and PCL displayed less than 1% weight loss. These findings align with the supposition posited by previous researchers [1] that temperature significantly affects the degradation process. In addition, Volova et al. (2010) [4] reported that PHB and PHBV films experienced 46% mass loss after being submerged in seawater at 29 °C for 160 days, with complete sample degradation occurring after 350 days, as reported in a similar study [1].

PLA degradation is a complex phenomenon involving several steps. Initially, its polymer chains undergo hydrolysis, which breaks down the bonds between the individual monomers that constitute the polymer. Water and other solutions permeate the PLA matrix and initiate the hydrolysis reaction, causing the disintegration of the polymer chains. Once the polymer chains are broken down into smaller fragments, the resulting lactic acid oligomers are metabolised by microorganisms present in the environment. This process involves the enzymatic degradation of lactic acid oligomers by microorganisms, which consume the fragments and use them as a source of carbon and energy [26,27].

In an oxygen-rich environment, the biodegradable polymers PLA and PCL undergo degradation primarily into CO_2_, CO, water and short-chain acids. However, blending these polymers with PHB can have a significant impact on the degradation process. The addition of PHB promotes the initial degradation of crotonic acid (2-butenoic acid), followed by the degradation of the aforementioned products [16]. This occurs due to the presence of microbial communities that preferentially consume the crotonic acid in the blended polymer mixture. As the crotonic acid is consumed, the degradation of the other components is accelerated, leading to the ultimate breakdown of the polymer blend into its constituent components [28].

As the average temperature and microorganism density in seawater are relatively low, the biodegradation of PLA in this environment occurs slowly. In a previous degradation study [6] using thin films of PLA (0.05 mm of thickness), after 10 weeks, samples in a real seawater solution at 25 °C showed an insignificant change in the average molar mass (M_w_). Additionally, PLA films of about 0.32 mm thickness presented no mass loss in a seawater solution, even after a year of exposure. Nevertheless, studies comparing the degradation of PCL and PHB films (thickness of 0.1 mm) at 25 °C [29] in seawater reported weight loss of 100 and 41%, respectively, after 4 weeks. The same study conducted with freshwater presented weight loss of only 25 and 9% for PCL and PHB, respectively, after 10 weeks. confirming the significance of the biopolymer type, geometry of the samples, and also the conditions of the environment, such as temperature and type and concentrations of marine microbes in the seawater [1,7,29].

Similarly, research carried out in a marine pelagic environment using 25 mg extruded films of blends produced with PBS, PHB, PLA and PCL reported promising relative biodegradation (wt.%) over a period of 56 days [15]. However, when comparing the relative biodegradation of PHB/PCL (60/40) samples reported in the previous study, which demonstrated approximately 15 wt.% weight loss in 25 mg films, with the blend 50/0/50 (PHB/PLA/PCL) used in our current research involving injection-moulded parts, we observed a higher loss of weight. These findings suggest that our approach, which utilises the industrial production method of injection moulding to create more densely packed components, yields more rapid biodegradation in the produced blends compared to extruded components.

Some studies reported delayed disintegration of the blends due to the selectivity of degradation of specific polymeric chains. The significant effect of hydrolysis on these polymeric chains was reported as being responsible for changes in the mechanical properties, resulting in softening of the blends, including PLA samples [7].

Aliphatic polyesters are known to biodegrade via the hydrolytic breakdown of their ester bonds [30]. The by-products of hydrolytic biodegradation lead to a more acidic environment, which raises a concern for implantable devices, as a common response would be local tissue inflammation. To lessen the acidity, researchers reported adding basic salts in the media or within the polyester samples [31,32].

The chemical hydrolysis rate increases in amorphous areas around the polymers due to the facilitated diffusion of water into the polymeric matrix compared to semi- and crystalline polymers, which present more organised structures causing diffusion limitation, even at temperatures higher than the Tg. Moreover, the diffusion of polymers with Tg values lower or similar to the temperature used in the degradation studies is primarily governed by the amorphous area of the polymeric matrix, where chemical hydrolysis is dominant [10,31].

According to previous research, the amount of PCL added to a polymeric matrix can be altered to control the rate at which the copolymers degrade [33]. These studies explored the addition and crosslinking of different PCL contents to fine-tune the degradation behaviour of copolymers, providing a greater degree of control over the release of therapeutic agents or other functional components. This has significant implications for the development of biomaterials that require precise degradation profiles as well as for the engineering of implantable medical devices and drug delivery systems [31,34].

### 3.2. Mechanical Properties

The PHB/PLA/PCL-based specimens were tested at a low strain rate of 5 mm·min^−1^, and the results are shown in Figure 2 after 0 and 56 days of seawater degradation. The tensile properties were acquired using equipment software after each test. However, to confirm the accuracy of this information, sample data were used to calculate the mechanical properties of the polymer blends.

The tensile results of neat PLA presented a higher yield stress resistance due to its higher intrinsic brittleness, while the remaining samples, including PHB, PCL, and blends of PHB, PLA, and PCL, presented a notable increase in the ultimate strain, evidenced by the higher flexibility of these materials. Similar results were reported in previous studies using PLA and PHB blends [35]. Notwithstanding, previous studies on the toughness of PLA [36] reported that the tensile strength and elastic modulus of ternary blends may differ slightly compared with neat PLA, implying that these mechanical properties are mainly influenced by the content of the PLA phase in the blends. And because PLA is more susceptible to thermal and hydrolytic degradation, a reduction in molecular weight due to the degradation process might lead to lower mechanical properties in the final material [37,38]. This is evidenced in Figure 2b, where all samples became more brittle after 56 days of seawater degradation.

Neat PLA is quite brittle when compared to PHB and PCL, as observed from the high stress and low strain relationship seen in Figure 2; it has been reported to have a low yield elongation of about 7% [39], and a modulus of about 3.4 GPa [11], whereas its tensile strength was found to be approximately 70 MPa, as seen in Figure 3 and supported by previous investigations [12]. These properties may limit its application in certain fields that require materials with higher ductility and toughness, such as flexible medical devices, or some packaging [40]. Previous investigations focusing on PLA/PCL blends have documented a reduction in sample stiffness with increasing PCL content, while conversely observing an increase in the elongation at break within the same samples. Furthermore, these studies reported a notable enhancement in impact strength with higher PCL concentrations. Specifically, the impact strength exhibited an improvement of approximately 200% with the incorporation of 30 wt.% PCL and a significant increase of approximately 350% when 40 wt.% PCL was added [41].

Tensile strength is an important mechanical property of polymers because it reflects the ability of any material to withstand pulling or stretching forces, which is commonly assessed for applications such as medical devices and packaging. The results provided in Figure 3 corroborate that PLA stands as the polymer with the highest tensile strength resistance in this study, and blends containing PLA also presented better performance in the tensile tests, as expected. As one of the drawbacks related to PLA is the low intrinsic elongation at break, the addition of PHB and PCL to tune the overall flexibility of blends is a promising solution for brittle samples, such as PLA. Previous investigations added different amounts of PCL to improve the ductile properties of PLA, which works similarly as plasticizers; however, the usual outcome is lower tensile strength [42,43]. As aforementioned, unblended PLA samples presented tensile strength results of 67 MPa, while with the addition of PHB and PCL, values of about 18 MPa (a decrease of 73.1%) and 25.5 MPa (a decrease of 61.9%) were obtained for the samples 50/25/25 and 30/50/20 (wt.% PHB/PLA/PCL), respectively, for samples not exposed to the seawater degradation environment. Furthermore, seawater exposure can lead to the absorption of the seawater solution by the polymeric blends, which may vary according to the hydrophilic nature of each individual polymer. This can cause swelling, softening, and plasticization of the blends, potentially leading to a decrease in mechanical properties such as tensile strength and stiffness, as reported in previous investigations [44]. Additionally, exposure to high temperatures, UV radiation, and mechanical stress can accelerate the degradation process.

### 3.3. Infrared Analysis

To better understand the main differences in the chemical structure and type of bonding, all samples were analysed at different times of seawater degradation after 0, 14, 28, and 56 days. A previous study suggested that PLA degradation is primarily driven by the generation of acetaldehyde and lactide monomers, although small amounts of carbon dioxide and carbon monoxide have also been detected [45]. The main degradation product of PHB has been identified as crotonic acid (2-butenoic acid) by numerous researchers. This assertion is supported by the characteristic peaks of the compound at 3056, 1722, and 1152 cm^−1^, which are attributed to C–H, C=O, and C–O stretching vibrations, respectively [46]. In addition, other studies have indicated the release of ester groups during PHB degradation [47]. PLA hydrolysis degradation and formation of oligomeric lactic acids (OLAs) were supported by the development of a double peak related to CH_3_ stretching at 1454 cm^−1^ and another double peak at 1743 cm^−1^ related to the C=O stretching vibrations [23,44]. The degradation of PCL is characterised by the evolution of ε-caprolactone (approximately 1736 cm^−1^) and 5-hexenoic acid and its di- and trimers (approximately 3475 cm^−1^); however, carbon dioxide and some traces of carbon monoxide have also been previously identified [45]. All FTIR peaks are shown in Appendix A.

The results presented in Table 2 show the ratio of the peak areas of the neat polymers PHB, PLA, and PCL, as well as their respective blends, after 0, 14, 28, and 56 days of seawater degradation. The ratios were determined based on the axial deformation of the bands A_1_ = 3050–2800 cm^−1^ (–CH_3_, –CH_2_), A_2_ = 1840–1600 cm^−1^ (–C=O), and A_3_ = 1250–1000 cm^−1^ (–C–O), which correspond to the specific chemical groups in the polymers of this study. Additionally, these results revealed that at the longest seawater degradation time (56 days), the neat PLA and PHB polymers exhibited a reduction in the peak ratio of areas A_2_/A_1_ and A_3_/A_1_ related to the C=O and C–O stretching absorption bands in relation to the nonpolar peaks (–CH_3_, –CH_2_), suggesting the evolution of the degradation of these polymeric chains, as confirmed by previous investigations [23,44]. The ratio of peak areas of PCL infrared samples remained slightly the same, suggesting that seawater degradation did not affect the chemical structure of the polymer over the time of these investigations, and thus it also affected blends containing PCL.

For the polymer blends, the ratios of the peak areas varied depending on the blend ratio and degradation time. For the 50/50/0 blend, the ratio of the peak areas for A2/A1 decreased after 28 days but increased again at 56 days, indicating a complex degradation pattern. The same behaviour was observed for the peak ratios of A3/A1. For the 50/0/50 blend, the ratio of the peak areas for A2/A1 and A3/A1 increased over time, indicating that the degradation of both PHB and PCL occurred preferentially in the non-polar structures. For 50/25/25 and 30/50/25, the ratios of the peak areas remained relatively constant over the 56-day period, suggesting that the degradation of these blends was not significantly affected by seawater.

### 3.4. Thermal Transitions and Degradation of PHB/PLA/PCL Blends

All neat polymers investigated (PHB, PLA, PCL) degrade in a one-step stage with typical representative temperatures T_5_, T_10_, T_max_ and the residual weight summarised in Table 3, for both neat polymers and blends. Blends containing either two or three polymers exhibited more than a one-step degradation process, and it is worth noting that the degradation of PLA is slightly delayed within the blends compared to neat PLA, as can be seen by how the T_5_ and T_10_ change in different blends [11,45]. Emphasising the blend with the lowest amount of wt.% PHB (30/50/20), presenting the thermal degradation of 312.3 °C (T_5_).

Previous studies found that the presence of PCL in similar blends delayed the degradation of PLA, meaning that PCL promotes the thermal stability of the blends. Studies using blends of 50/50 and 70/30 (wt.% PCL/PLA) compositions reported an increase in the characteristic decomposition peak temperature of PLA to 325 °C for 50/50 blends and 334 °C for 70/30 blends [11,46]. Similar results were found using blends of PLA and PCL with only 30 wt.% PCL. The latter suggested that the immiscibility between PLA and PCL is evidenced by the presence of a two-step degradation process and that the amount of weight loss is proportional to the polymer content in each blend during degradation [48,49,50]. According to Arrieta et al., ternary blends such as PHB/PLA/PCL tend to degrade in a three-step process, where each process corresponds to the individual polymer and the mass loss for each step is proportional to the polymeric content in the blend; the same behaviour is observed in the blends in this study, as reported in the blends 50/25/25 and 30/50/20 (wt.% PHB/PLA/PCL) in Figure 4 [51].

The DSC thermograms for blends containing both PLA and PCL, such as 50/25/25 and 30/50/20 (wt.% PHB/PLA/PCL) revealed that the melting peak of PCL overlaps with the T_g_ of PLA at approximately 60 °C. For this reason, conventional DSC is unable to detect fine variations in the T_g_ of PLA in blends containing PCL. To mitigate this challenge, DMTA analysis of samples was used to detect the T_g_ points of PHB, PLA, and PCL; the values found are presented in Table 4. Due to the diluting effect in the blend, the cold crystallisation process is visible as a slightly broad peak at the same location as in the neat PLA. This shows that PCL has little effect on the crystal structure of PLA, strongly supporting their poor miscibility [11]. The DSC thermograms of neat polymers and their respective blends are shown in Appendix A.

Table 4 shows that the glass transition temperature (T_g_) of PLA is around 67.2 °C after 56 days of seawater degradation, which is slightly higher than its original T_g_ of 66.7 °C before degradation, similar effect happened to the T_g_ of PCL, whereas the T_g_ of PHB increased from 55.3 °C to 62.9 °C after the same period. This suggests that seawater degradation did not have a significant effect on the T_g_ of PLA and PCL, but caused the PHB chains to become more rigid, possibly due to the formation of cross-links between the chains [48,51].

Furthermore, the T_m_ values of both PLA and PHB remained relatively constant, indicating that seawater degradation did not significantly affect the crystallinity of these polymers. In contrast, the T_m_ of PCL increased from 61.3 °C to 65.0 °C after degradation, and it also increased its crystallinity from 45.4% to 60.7% after 56 days of seawater degradation, due to the amorphous regions of the polymer being more susceptible to hydrolysis degradation.

The data also demonstrated that the thermal properties of the polymeric blends were similar to those of their respective neat polymers, varying according to the ratio of each polymer within the blend. For instance, the T_g_ of the 50/50/0 blend of PLA and PHB was similar to that of neat PHB, but slightly lower than that of neat PLA. Seawater degradation also had a similar effect on the thermal properties of the blends, with the crystallinity indices of neat PLA and PHB polymers remaining slightly constant after degradation, whereas the crystallinity of PCL increased. A similar effect was observed for the respective blends, which varied according to their polymer ratios.

### 3.5. Morphology of the Samples

The examination of fractured specimens after tensile testing revealed the brittle fracture of samples of neat PLA and PHB at 0 and after 56 days of seawater degradation. Both polymers presented lower strain before failure in the stress–strain curves, as reported in Figure 2. A ductile fracture preceded by significant plastic deformation was observed, associated with extensive stress whitening that occurred throughout nearly the entire gauge zone of the specimen, which diminished in the same specimens after 56 d of seawater degradation. This was observed in the blends containing 50 wt.% of PLA, such as both 50/50/0 and 30/50/20 formulations, as shown in Figure 5. Furthermore, it was observed that blends with weight ratios of 50/0/50 and 50/25/25 (wt.% PHB/PLA/PCL) exhibit a greater occurrence of white spots, which are indicative of surface degradation accompanied by a loss of ductility and transition to brittle fracture behaviour [35], after 56 days of seawater degradation.

SEM photomicrographs revealed the formation of cracks and fissures on the surfaces of the PLA, which is also a typical indication of polymer degradation [52,53]. The formation of cracks on the surface of PLA samples is an important factor in promoting the diffusion of OLAs into the degradation environment [54]. These cracks increased the surface area of PLA, which was exposed to an environment that promoted degradation, thereby facilitating the diffusion of OLAs and other degradation products. The increased diffusion of OLAs from PLA also promotes local bacterial activity, as the microorganisms present in the environment can more readily access OLAs and use them as a source of carbon and energy. This localised bacterial activity can further accelerate the degradation of PLA, leading to a quicker breakdown of the polymer chains and the release of carbon dioxide and other by-products [24].

As depicted in Figure 6, the SEM analysis of the studied blends revealed uneven surface topography and disordered internal structures, which are indicative of phase separation within the blend. These observations are consistent with previous investigations and with the behaviour of immiscible polymer blends [55]. This is in contrast to the smooth and uniform fracture surfaces characteristic of neat PLA and PHB, which present brittle fracture behaviour and might be classified as rigid fillers in the blends, whereas the presence of PCL improves the overall ductility of the blends, as evidenced by SEM photomicrographs and mechanical properties from previous investigations [56,57,58,59].

In previous studies on ternary blends of PHB, PLA, and PCL, researchers reported that neat PLA has a brittle fracture with a typical smooth surface presenting several crack fronts [11]. Regarding PLA/PHB blends (60/40 wt.%), SEM photomicrographs also showed a morphology indicating phase separation between the polymers, and were confirmed by thermal analysis [35]. In other studies, the immiscibility of the blends was evidenced by the bead-like shape of the blends containing PCL, with the PCL dispersed phase containing spherical shapes ranging from 1 to 10 µm [39,60].

## 4. Conclusions

The present investigation utilised a methodology to fabricate and assess blends composed of biodegradable materials such as PHB, PLA, and PCL, resulting in the production of materials with customised mechanical properties. The evaluation of seawater degradation mechanisms revealed the occurrence of surface erosion and bulk degradation in the blends, where the weight variance was initially positive in certain blends, indicating the presence of bulk degradation with the formation of cracks and fractures that permitted the permeation of fluids and salts, subsequently resulting in an increase in weight. However, this effect tends to increase the degradation rate over time by augmenting the surface area of degradation and the local autocatalysis effect. These findings have significant implications for understanding the degradation behaviour of these polymers in a seawater environment.

In light of the findings of this study, the development of tailored mechanical properties for PHB/PLA/PCL blends is suggested by adjusting the content of each biopolyester. The tensile results indicated that all samples exhibited increased brittleness after 56 days of seawater degradation, with PLA being the most rigid polymer among the investigated materials. The blending process of the samples in this study with PCL promoted the flexibility of the blends, especially the ones with the highest content of PLA. As corroborated by previous studies, the addition of different amounts of PCL causes a decrease in the stiffness of PLA-based samples by improving their elongation at break and impact strength properties, which facilitates their use in packaging and medical device applications. Furthermore, neat polymers underwent degradation in a one-step stage, whereas blends containing two or three polymers underwent a multistep degradation process, indicating a degree of immiscibility between the polymers. Morphological analysis of the samples revealed brittle fracture in PLA and PHB and ductile fracture preceded by significant plastic deformation in blends containing 50 wt.% of PLA, such as 50/50/0 and 30/50/20 formulations, accompanied by extensive stress whitening across the gauge zone. In addition, blends with weight ratios of 50/0/50 and 50/25/25 (wt.% PHB/PLA/PCL) exhibited a higher frequency of white spots, indicating surface degradation, loss of ductility, and transition to brittle fracture behaviour. Based on the findings of this study, it is recommended to develop a biopolyester blend with a high PCL content to promote ductility, in combination with a suitable ratio of PHB and PLA to achieve the desired mechanical properties.

## Figures and Tables

**Figure 1 polymers-15-02874-f001:**
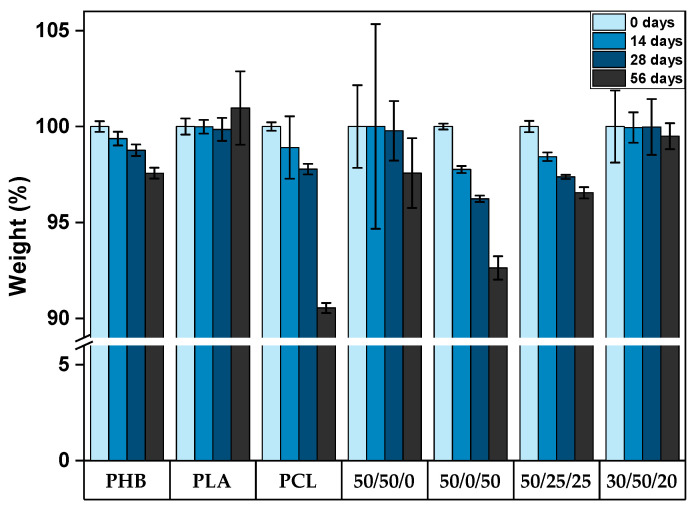
Weight variance (%) of PHB/PLA/PCL and their respective blends with different compositions (wt.% PHB/PLA/PCL) after 0, 14, 28 and 56 days of seawater degradation.

**Figure 2 polymers-15-02874-f002:**
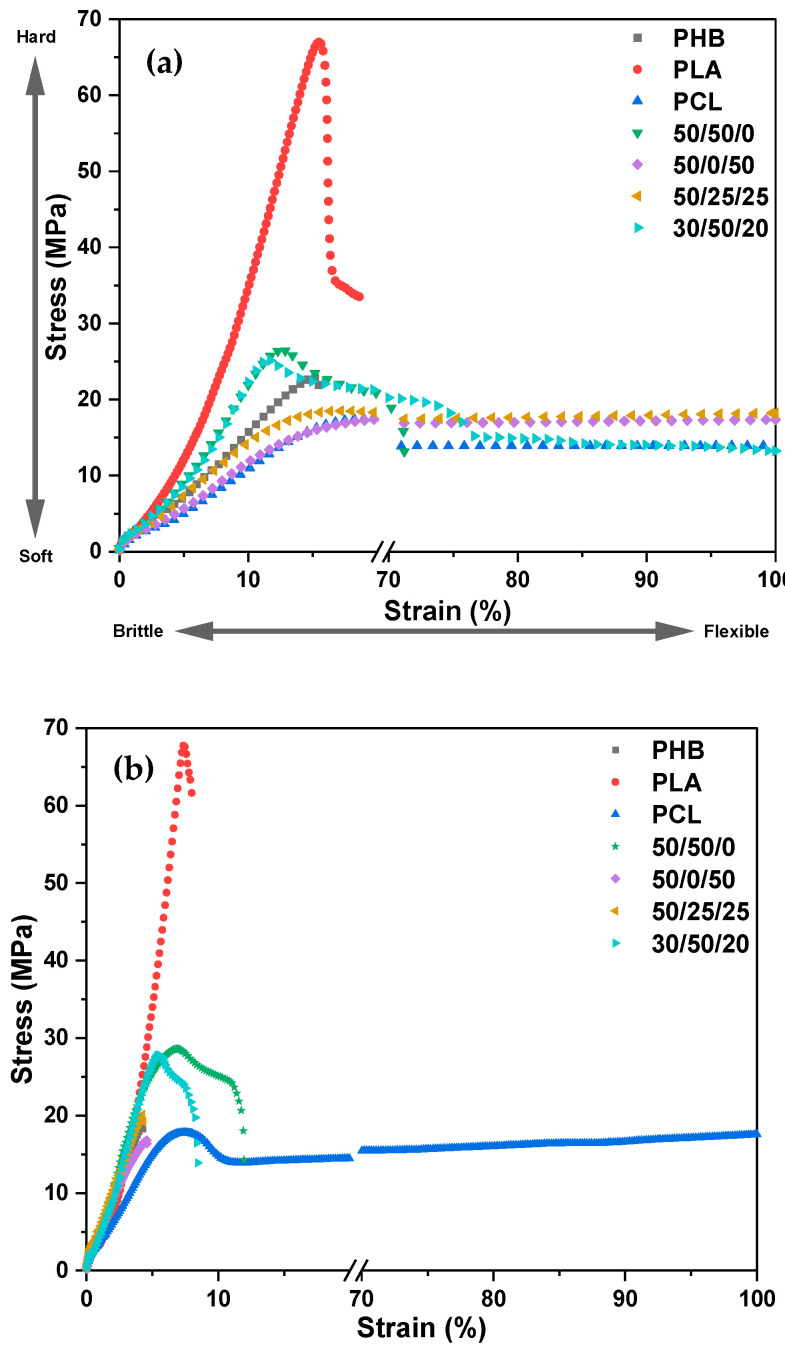
Stress-Strain curves of PHB/PLA/PCL and their respective blends with different compositions (wt.% PHB/PLA/PCL) after 0 days in (**a**), and after 56 days of seawater degradation, in (**b**). Individual curves of Stress vs. Strain after 0 and 56 days for each material are plotted in Appendix A.

**Figure 3 polymers-15-02874-f003:**
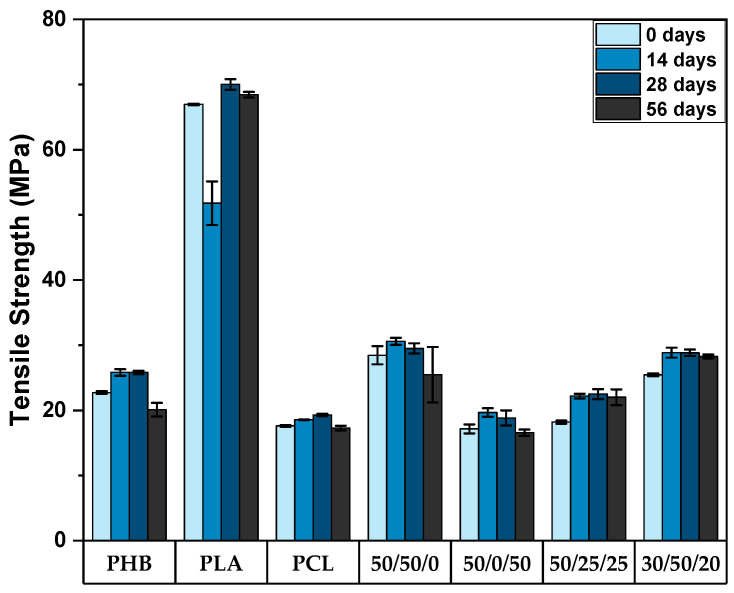
Tensile strength of PHB/PLA/PCL and their respective blends with different compositions (wt.% PHB/PLA/PCL) after 0, 14, 28 and 56 days of seawater degradation.

**Figure 4 polymers-15-02874-f004:**
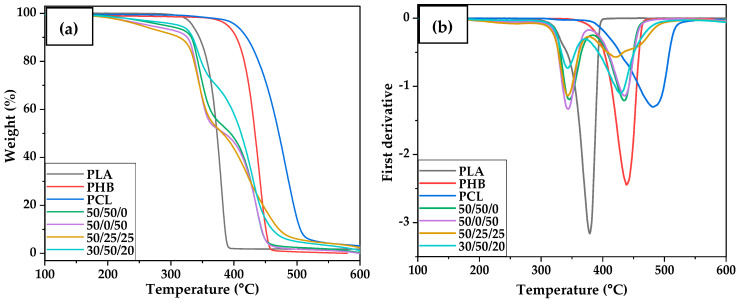
Comparative plot of the thermal degradation of PHB, PLA, and PCL samples and their respective ternary blends with different compositions (wt.% PHB/PLA/PCL), (**a**) TGA (thermograms), and (**b**) DTG (first derivative curves).

**Figure 5 polymers-15-02874-f005:**
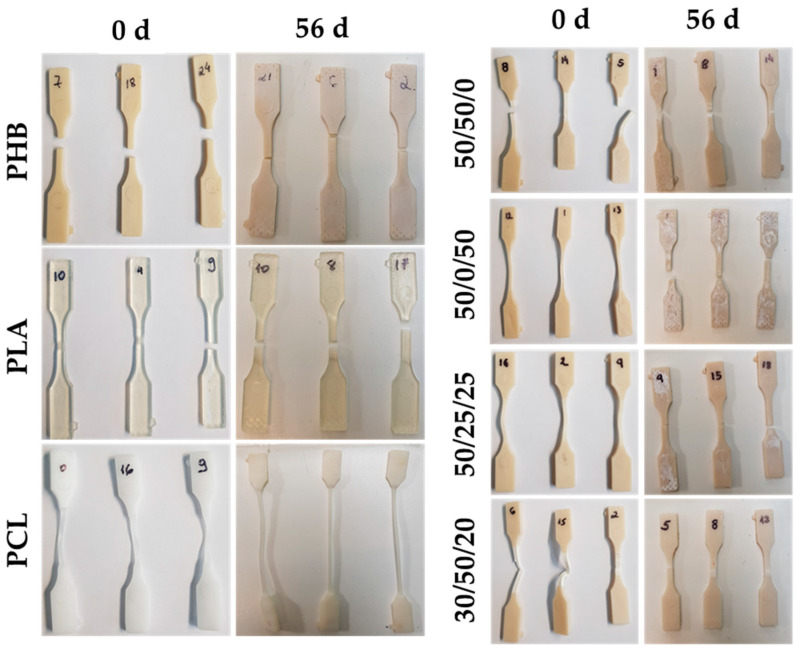
Comparative of the different morphologies of PHB, PLA, and PCL samples and their respective ternary blends with different compositions (wt.% PHB/PLA/PCL), after tensile testing at 0 and after 56 days of seawater degradation.

**Figure 6 polymers-15-02874-f006:**
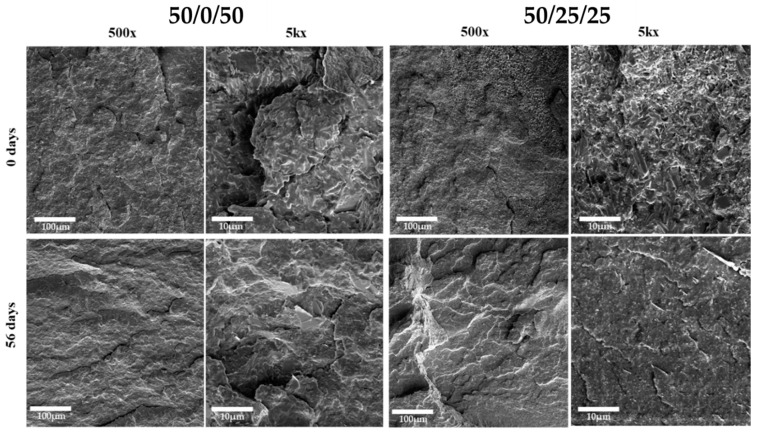
SEM photomicrographs of blends 50/0/50 (**left**) and 50/25/25 (**right**) (wt.% PHB/PLA/PCL) at 0 and after 56 days of seawater degradation.

**Table 1 polymers-15-02874-t001:** Injection moulding process parameters of PHB, PLA, and PCL and their respective blends with different compositions (wt.% PHB/PLA/PCL).

Sample	Temperature (°C)			Shot Size (mm)	Cooling		Pressure (Bar)	
	Zone 1	Zone 2	Zone 3		T (°C)	t (s)	1st	2nd
PHB	185	175	170	35	50	25	40	30
PLA	210	200	180	40	35	35	80	70
PCL	170	160	150	40	35	35	100	90
50/50/0	160	150	140	37	35	35	90	85
50/0/50	150	140	130	36	35	35	70	65
50/25/25	150	140	130	36	35	35	70	65
30/50/20	160	150	140	36	35	35	90	85

**Table 2 polymers-15-02874-t002:** Ratio of peak areas of FTIR spectra of neat polymers and their respective blends (wt.% PHB/PLA/PCL) at 0, 14, 28, and 56 days.

		Days			
	Ratio	0	14	28	56
PHB	(A_2_/A_1_)	5.17	3.91	14.13	3.78
	(A_3_/A_1_)	5.28	3.49	15.27	3.66
PLA	(A_2_/A_1_)	11.68	8.37	10.52	9.09
	(A_3_/A_1_)	31.00	20.48	27.24	23.19
PCL	(A_2_/A_1_)	1.46	1.41	1.44	1.33
	(A_3_/A_1_)	1.65	2.24	1.94	2.02
50/50/0	(A_2_/A_1_)	5.35	5.19	3.18	5.63
	(A_3_/A_1_)	10.05	8.56	4.76	9.35
50/0/50	(A_2_/A_1_)	2.35	2.08	2.22	4.05
	(A_3_/A_1_)	2.43	2.15	1.88	4.72
50/25/25	(A_2_/A_1_)	3.58	3.03	3.37	3.23
	(A_3_/A_1_)	4.44	3.96	3.96	3.96
30/50/25	(A_2_/A_1_)	4.03	4.46	4.13	4.49
	(A_3_/A_1_)	5.58	6.48	5.84	7.10

Axial deformation of bands: A_1_ = 3050–2800 cm^−1^ (–CH_3_, –CH_2_); A_2_ = 1840–1600 cm^−1^ (–C=O); A_3_ = 1250–1000 cm^−1^ (–C–O).

**Table 3 polymers-15-02874-t003:** Thermal degradation properties of PHB/PLA/PCL blends obtained by TG/DTG characterisation. Curves of TG/DTG and the respective deconvolution of each thermal degradation are represented in Appendix A.

Material	T_5_ (°C)	T_10_ (°C)	T_max_ (°C)			Residual Weight (%)
			PLA	PHB	PCL	
PHB	389.9	403.7	-	435.2 ± 0.1	-	0.26 ± 0.1
PLA	247.1	292.2	339.2 ± 0.0	-	-	1.82 ± 0.1
PCL	403.7	420.2	-	-	472.0 ± 0.1	4.26 ± 0.2
50/50/0	290.6	332.4	348.0 ± 0.1	429.3 ± 0.1	-	1.86 ± 0.2
50/0/50	265.3	326.5	-	344.2 ± 0.1	430.4 ± 0.1	1.20 ± 0.1
50/25/25	252.2	313.3	343.6 ± 0.1	421.5 ± 0.5	464.5 ± 0.5	4.21 ± 0.2
30/50/20	312.3	334.4	343.5 ± 0.1	380.6 ± 0.4	426.1 ± 0.1	3.27 ± 0.3

T_5_, T_10_ and T_max_ being the temperatures at 5%, 10% and maximum weight loss rate.

**Table 4 polymers-15-02874-t004:** Differential scanning calorimetry (1^st^ heating curves) for neat polymers and their respective blends (wt.% PHB/PLA/PCL) after 0 and 56 days of seawater degradation.

Sample Name	d	T_g_ (°C)	T_m_ (°C)	T_c_ (°C)	ΔH_c_ (J/g)	ΔH_m_ (J/g)	Crystallinity Index (%)
PHB	0	55.3	167.0	110.6	−76.8	67.2	46.0
	56	62.9	166.8	111.2	−72.2	65.2	44.7
PLA	0	66.7	149.9	59.2	−5.5	6.8	7.3
	56	67.2	150.1	59.4	−4.9	7.0	7.5
PCL	0	−43.1	61.3	28.1	−65.3	61.3	45.4
	56	−41.7	65.0	28.7	−71.0	82.2	60.9
50/50/0	0	53.8	PLA (141.8), PHB (169.8)	PLA (97.2), PHB (106.4)	PLA (−1.3), PHB (−32.2)	PLA (10.8), PHB (27.4)	PLA (11.5), PHB (18.8)
	56	55.7	PLA (145.2), PHB (169.8)	PLA (97.9), PHB (107.5)	PLA (−1.8), PHB (−20.0)	PLA (9.9), PHB (29.7)	PLA (10.6), PHB (20.3)
50/0/50	0	29.4	PCL (58.3), PHB (169.0)	PCL (29.4), PHB (105.6)	PCL (−35.6), PHB (−36.7)	PCL (34.4), PHB (34.9)	PCL (25.5), PHB (23.9)
	56	29.8	PCL (63.7), PHB (171.4)	PCL (29.8), PHB (107.1)	PCL (−34.2), PHB (−34.9)	PCL (39.7), PHB (34.3)	PCL (29.4), PHB (23.5)
50/25/25	0	58.8	PCL (58.8), PLA (143.5), PHB (168.8)	PCL (29.4), PHB (106.2)	PCL (−18.6), PHB (−29.0)	PCL (15.8), PLA (4.0), PHB (27.1)	PCL (11.7), PLA (4.3), PHB (18.6)
	56	60.1	PCL (60.1), PLA (143.8), PHB (168.7)	PCL (29.8), PHB (106.6)	PCL (−21.1), PHB (−31.7)	PCL (26.6), PLA (4.5), PHB (30.4)	PCL (19.7), PLA (4.8), PHB (20.8)
30/50/20	0	59.8	PCL (59.5), PLA (144.5), PHB (172.8)	PCL (30.2), PHB (105.0)	PCL (−11.4), PHB (−16.8)	PCL (9.3), PLA (9.2), PHB (13.7)	PCL (6.9), PLA (9.8), PHB (9.4)
	56	61.4	PCL (61.4), PLA (146.4), PHB (171.4)	PCL (30.8), PHB (105.6)	PCL (−15.1), PHB (−21.1)	PCL (21.7), PLA (11.8), PHB (17.8)	PCL (16.1), PLA (12.6), PHB (12.2)

## Data Availability

The data that support the findings of this study are available from the corresponding author upon reasonable request.

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
