# Peer review of "Designing Sustainable Polymer Blends: Tailoring Mechanical Properties and Degradation Behaviour in PHB/PLA/PCL Blends in a Seawater Environment"

_polymers, 2023, doi:10.3390/polym15132874_

Round 1
Reviewer 1 Report
The issue of biodegradable polymers is very topical and important nowadays. Obtaining products that meet specific performance criteria and meet the requirements of biodegradation is very difficult and requires many tests and tests. The authors address this topic in their manuscript.
The presented introduction of the work and the analysis of the literature well describe the purpose of the work. The cited literature is up-to-date and well-chosen to the subject of the work.
The methodology of the experiment was correctly adopted.
The description of the tables and figures is correct.
The obtained polymer mixtures have very interesting results.
Admittedly, the extrusion process was not analysed.
Conclusions are correctly formulated.
Details below:
The aim of the authors was to assess the properties of PLA/PHB polymer mixtures. The authors have shown that these types of mixtures are feasible. They obtained them in the extrusion process. This process ensures good mixing
The discussed issue is very topical, especially since the proposed sets of obtained materials will be intended for applications in medicine and packaging. In this case, there is a certain contradiction between biodegradation and performance. On the one hand, the material is to be biodegradable, on the other hand, it is to be durable and resistant to the environment so that it meets the relevant requirements. It is true that there are similar publications in this area, but this issue is still very valid.
The presented manuscript complements the information on the properties of PLA/PHB mixtures. There is a bit missing description of the extrusion process and the phenomena that occur during extrusion, but this does not diminish the value of the work.
Currently, due to the great interest in the problem of environmental protection, a large number of publications related to modifications of the most popular biodegradable polymers appear. The work fits into this area and complements it.
The authors should supplement the information on the extrusion process, there is no information related to the measurement methodology. It would be good to provide measurement uncertainty or at least standard deviations. It is worth supplementing the work with the simplest measurements of biodegradation, weight loss over time, or changes in properties. That would greatly affect its quality.
Good literature is correct and up-to-date, it gives a good background to the problem posed at work. There are older items in it, but they are still valid. It is sufficient.
Figures and tables are correctly described, descriptions are also in the main text. The final results could have been slightly modified to emphasize the value of the results obtained.
Reviewer 2 Report
Manuscript Number: polymers-2421721
Title: Designing Sustainable Polymer Blends: Tailoring Mechanical 2 Properties and Degradation Behaviour in PHB/PLA/PCL Blends 3 in a Seawater Environment
This manuscript concerns the production and thermal/mechanical/morphological characterization of ternary blends of biodegradable polyesters, PLA/PCL/PHB, for packaging applications and biomedical devices. A study on the degradation of blends produced in sea water under controlled conditions was also conducted monitoring the changes of weight of tensile specimens and their mechanical properties during the degradation test.
General comment
I suggest reviewing English in some parts.
Several research papers on PLA/PCL/PHB ternary blends have already been published (eg doi: 10.3390/polym10010003; doi: 599 10.3390/polym10010003 and others). So, in the introduction the authors should highlight the innovative aspects of this work. Furthermore, two bioplastics such as PCL and PHB, biodegradable in sea water, were blended with PLA which is well known to be non-biodegradable in sea water. It is not clear why this ternary mixture was chosen if a biodegradation test in seawater was carried out on this ternary mixture. In the introduction, the authors focus mainly on the use of bioplastics in the production of biomedical devices and the biodegradation mechanisms involved in these applications. Therefore, it is not clear why biodegradation in sea water was evaluated if no applications in such an environment are envisaged. Finally, in the introduction the authors assert that bioplastics can solve the problem of plastic pollution in the sea. I don't fully agree with that statement, the solution to this problem is not that simple. We bear in mind that bioplastics, by their nature and properties, will only be able to replace traditional plastics to a small extent, and the problem of plastic pollution at sea requires policies aimed at a more efficient collection and recycling systems for plastics as well as more respectful human behavior towards the environment. Presently, the main benefits of bioplastics are linked to waste management (including collection and recycling of organic waste). The bioplastics can play an important role in those applications where their biodegradation into non-toxic compounds at the end of use is an added value, thus avoiding the need to collect and dispose of them and their possible accumulation in the environment as happens with traditional plastics.
So, highlight the innovative aspects of the present work, the reason for characterizing this ternary mixture containing PLA, no-sea-water biodegradable, in terms of biodegradation in the sea, and the applications where the use of bioplastics leads to benefits respect to traditional plastics.
Specific comments
In the abstract
1. I suggest reporting also the most significant quantitative results such as optimal compositions of the mixtures produced, mechanical properties, rate of biodegradation in sea water, ...
In the Introduction
2. Page 2 line 49 and 56: the authors assert that PLA is not biodegradable in the sea but then cite a work which shows that the mechanical action of the waves promotes the degradation of PLA; perhaps it is more correct to talk about mechanical fragmentation.
3. I suggest reporting the aim of the work at the end of the Introduction. At page 2, the sentences in the lines 89-93 are a repeat of lines 67-72.
4. At page 8 lines 63-66: since the article deals with blending of polymers with PLA, it is not clear why the example of composite of PLA and halloysite nanotubes.
Materials and methods
5. Page 3 lines 97, 99, 101: please report the conditions (Temperature and the load) used for the MFI.
Results and Discussion
6. Page 5 line 224: please report the environments where the bulk degradation of PLA has been studied, ref. 22, 23.
7. Page 5 lines 224-225: the authors assert that a bulk degradation of PLA occurred but figure 1 shows that the weight of the PLA samples remained unchanged or even increased after 56 days, probably due to the absorption of salt water which released salts inside by drying before weighing.
8. Page 5 line 232: I suggest replacing “notable” with “not negligible”, considering also that the temperature of simulated sea water is high (30°C), 2,5% of weight loss after 2 months is not so high.
9. When the authors talk about the biodegradation mechanisms of PLA, I suggest highlighting the importance of the temperature required for the hydrolysis of the PA chains to occur at a detectable rate.
10. Page 7 line 274, take out of parenthesis “at 25°C”.
11. Page 7 line 304-305, I suggest removing these sentences, they are redundant,
12. About the results of mechanical properties: for a more immediate comparison between the various samples and the effect of time on their tensile properties, I suggest plotting, as in Figure 3, the young's modulus, the yield strength, and the elongation at break with the bars of the standard deviation.
13. the authors associate the increase in rigidity of the PLA and of the other samples after 56 days with the degradation of the material, with the reduction of the molecular weight but from the experimental data (figure 1) it does not appear that the PLA has degraded.
14. No comments on the effect of seawater exposure on the mechanical properties are reported, please report them.
15. Regarding the FTIR analysis: in the present work the samples during the degradation test in sea water were analyzed at room temperature. It is therefore not clear why in the discussion of the FTIR spectra, the results relative to the TGA-FTIR analysis carried out on PLA, PCL and PHB are reported, i.e. the products and the related functional groups that are formed during the thermal degradation of PLA, PCL and PHB. In my opinion, they are not useful for the discussion of spectra obtained at room temperature in the present work, am I wrong?
16. The authors use the trend of the area ratios of the characteristic peaks as an index of the degradation of the investigated polymers, concluding that sea water does not influence the chemical structure of PCL but that of PHB and PLA. Considering that the FTIR analysis is a surface analysis, could it be that in the case of PCL the hydrolysis products present on the surface have been removed by washing with water, leaving the surface only partially hydrolysed? it is also not clear why PCL blended with PHB (50-50) show an increase (about double) of the A2/A1 and A3/A1 ratios while pure PHB and PCL show decreases in both, better explain these trends.
17. Page 11 line 405, I suggest adding “slightly” before " delayed”.
18. In the footnote of Table 3: Tmax is the temperature of the maximum weight loss rate.
19. Page 11 lines 413-414: the authors assert that the ternary blends such as PHB/PLA/PCL tend to degrade in a three-step process, where each process corresponds to the individual polymer, and the mass loss for each step is proportional to the polymeric content in the blend; but for the blends 50/25/25 and 30/50/20 in Figure 4 the third peak attributable to PCL is not clearly visible; maybe by deconvolutions.
20. About the DSC results: for a more immediate comparison of DSC thermograms between the various samples at the initial time and after 56 days, I would report Figure S4 in the manuscript and Table 4 in the supplementary material.
21. At page 13 lines 452-454: the increase of crystallinity of PCL can be attributable to the fact that it is mainly the amorphous phase involved in the biodegradation with consequent increase of the percentage content of the crystallinity phase, as also asserted by the authors, and not to the effect of sea water to order the PCL chains.
22. At page 14 lines 482: the authors comment SEM images of PLA reported in other studies (Ref. 51, 52 23) without reporting the degradation environment and related conditions related to. Furthermore, the effect of the presence of OLAs in PLA on bacterial activity is mentioned. The usefulness of these references is not clear, given the non-addition of the OLAs, in the discussion of the morphology of the samples of the present work.
23. The authors compare the SEM photos of two blends with those of the pure polymers, but the latter were not reported, not even in the supplementary material, please report them. The authors mention the presence of fibrils, but they are not clearly visible, please highlight them in the photos with arrows.
Conclusions
In the manuscript the authors talk about surface degradation of PHB and PCL due to the constant weight loss rate, while in the conclusions about bulk degradation! I suggest reducing the conclusions by focusing on the effects of PCL on the stiffness of PLA to have a more ductile material, required for packaging and biomedical devices. The present study confirms the non-biodegradability of PLA in seawater and the biodegradability of PCL and PHB with the consequent changes in crystallinity and tensile properties.
Minor editing of English language required
Reviewer 3 Report
The present paper deals with the ternary biopolymer blends.
The paper is sound, however, several shortcomings need to be solved before publishing.
1) Abstract should present the results, including the numerical information and changes.
2) The idea of ternary blends is interesting, but the motivation to prepare these blends is missing. How the polymer concentration was chosen?
The received results can be expected. What is the contribution of the present research?
3) The innovation potential and the level above state of the art are unclear from the text.
4) The compatibility of the physical blends is not discussed. No compatibilizers have been used.
5) Low biodegradation of the current polymers in seawater is obvious in the chosen time frame and testing conditions. Why did the test end after 56 days?
Round 2
Reviewer 2 Report
Dear authors, I think that there are no innovative aspects in this work and the aim of the work, study of the effect of PHB and PCL on the biodegradability of PLA, is not supported by the results obtained. Therefore, as the present work has been set up, I think that it should not be considered in this form for publication in Polymers.
Minor editing of English language required.
Author Response
Dear reviewer, Thank you for your insightful review of our manuscript. We sincerely appreciate the time and effort you dedicated to assessing our work, and we are grateful for your valuable suggestions. Your feedback has significantly contributed to improving the quality of our manuscript. We acknowledge the importance of rigorous peer review to ensure the validity and clarity of scientific publications. Your constructive comments and recommendations have been instrumental in refining our research and enhancing the overall presentation of our findings. We appreciate your expertise and thoughtful evaluations of our work. Please let us know if there are any further suggestions or concerns that you would like us to address. We look forward to your final decision regarding our manuscript. Thank you once again for your time and expertise. Sincerely, The authors
Reviewer 3 Report
Now paper can be accepted for publication.
Author Response
Dear reviewer, Thank you for your insightful review of our manuscript. We sincerely appreciate the time and effort you dedicated to assessing our work, and we are grateful for your valuable suggestions. Your feedback has significantly contributed to improving the quality of our manuscript. We acknowledge the importance of rigorous peer review to ensure the validity and clarity of scientific publications. Your constructive comments and recommendations have been instrumental in refining our research and enhancing the overall presentation of our findings. We appreciate your expertise and thoughtful evaluations of our work. Sincerely, The authors